# SUG: Single-dataset Unified Generalization for 3D Point Cloud Classification

## Abstract

In recent years, research on zero-shot domain adaptation, namely Domain Generalization (DG), which aims to adapt a well-trained source domain model to unseen target domains without accessing any target sample, has been fast-growing in the 2D image tasks such as classification and object detection. However, its exploration on 3D point cloud data is still insufficient and challenged by more complex and uncertain cross-domain variances with irregular point data structures and uneven inter-class modality distribution. In this paper, different from previous 2D DG works, we focus on the 3D DG problem, and propose a Single-dataset Unified Generalization (SUG) framework that only leverages the source domain data to alleviate the unforeseen domain differences faced by the well-pretrained source model. Specifically, we first design a Multi-grained Sub-domain Alignment (MSA) method that can constrain the learned representations to be domain-agnostic and discriminative, by performing a multi-grained feature alignment process between the splitted sub-domains from the single source dataset. Then, a Sample-level Domain-aware Attention (SDA) strategy is presented, which can selectively enhance easy-to-adapt samples from different sub-domains according to the sample-level inter-domain distance, to avoid the negative transfer. Extensive experiments are conducted on three common 3D point cloud benchmarks. The experimental results demonstrate that SUG framework is effective to boost the model generalization ability for unseen target domains, even outperforming the existing unsupervised domain adaptation methods that have to access extensive target domain data, where we significantly improve classification accuracy by 7.7% on ModelNet-to-ScanNet setting and 2.3% on ShapeNet-to-ScanNet setting. Our code will be available.

## 1 Introduction

As a commonly-used data format describing the real world, point clouds-based representations preserve more geometric information residing in 3D scenes, and have become one of the most important data types for 3D scene perception and real applications such as robotics (Rusu et al., 2008; Rusu & Cousins, 2011), autonomous driving (Sun et al., 2020; Shi et al., 2020), and augmented and virtual reality (Tredinnick et al., 2016), giving a better understanding of the surrounding environment for machines. In recent years, point clouds-based vision tasks (Shi et al., 2020) have achieved great progress on the public benchmarks (Vishwanath et al., 2009; Chang et al., 2015; Dai et al., 2017), which largely owes to the fact that the collected point clouds are carefully annotated, sufficiently large, and low level noised. But in the real world, acquiring such data from a new target domain and manually labeling these extensive 3D data are highly dependent on professionals in this filed, which makes the data acquisition and annotation more difficult, labor-intensive, and time-consuming.

One effective solution to transfer the model from fully-labeled source domain to a new domain without extra human labor is Unsupervised Domain Adaptation (UDA) (Shen et al., 2022; Zou et al., 2021; Fan et al., 2022; Yang et al., 2021), whose purpose is to learn a more generalizable representation between the labeled source domain and unlabeled target domain, such that the model can be adapted to the data distribution of the target domain. For example, when point cloud data distribution from the target domain undergoes serious geometric variances (Shen et al., 2022), performing a correct source-to-target correspondence can boost the model's adaptability. Besides, GAST (Zou et al., 2021) learns a domain-shared representation for different semantic categories, while a vot-

ing reweighting method is designed (Fan et al., 2022) that can assign reliable target domain pseudo labels. However, these techniques are highly dependent on the accessibility of the target domain data, which is a strong assumption and prerequisite for the models running in an unprecedented circumstance, such as autonomous driving system and medical scenarios. Thus, it is meaningful and important to investigate the model's cross-domain generalization ability under the zero-shot target domain constraint, which derivates the task of **Domain Generalization (DG)** for 3D scenario.

However, achieving such zero-shot domain adaptation, *i.e.*, DG, is more challenging in 3D scenario mainly due to the following reasons. **(1) Unknown Domain-variance Challenge:** 3D point cloud data collected from different sensors or geospatial regions with different data distributions often present serious domain discrepancies. Due to the inaccessibility of the target domain data (or sensor), modeling of source-to-target domain variance is intangible. **(2) Uneven Domain Adaptation Challenge:** Considering that our goal is to learn a transferable representation that can be generalized to multiple target domains, a robust model needs to perform an even domain adaptation, rather than lean to fit the data distribution on one of the multiple target domains. But for 3D point cloud data with more complex sample-level modality variances, how to ensure an even model adaptation under the zero-shot target domains setting still remains challenging.

To tackle the above challenges, we study the typical DG problem in 3D scenario, and introduce a Singe-dataset Unified Generalization (SUG) framework for addressing the 3D point cloud generalization problem. We study a one-to-many domain generalization problem, where the model can be trained on only a single 3D dataset, and is required to be *simultaneously generalized to* **multiple target datasets**. Different from previous DG works in 2D scenarios (Shankar et al., 2018; Piratla et al., 2020; Chen et al., 2021), 3D point cloud data have a more irregular data structure and diverse data distribution within a single dataset, which provides the possibility to exploit the modality and sub-domain changes without accessing any target-domain datasets. To be specific, our SUG framework consists of a Multi-grained Sub-domain Alignment (MSA) method and a Sample-level Domain-aware Attention (SDA) strategy. To address the unknown domain-variance challenge, the MSA method first splits the selected single dataset into different sub-domains. And then, based on the splitted different sub-domains from a single dataset, the baseline model is constrained to simulate as many domain variances as possible from multi-grained features, so that the baseline model can learn multi-grained and multi-domains agnostic representations. To solve the uneven domain adaptation challenge, the SDA is developed, which assumes that the instances from different sub-domains often present different adaptation difficulties. Thus, we add sample-level constraints to the whole sub-domain alignment process according to the dynamically changing sample-level inter-domain distance, leading to an even inter-domain adaptation process.

We conduct extensive experiments on several common benchmarks (Qin et al., 2019) under the single-dataset DG scenario, which includes three sub-datasets and our experiments cover the following three scenarios: 1) **ModelNet-10→ShapeNet-10/ScanNet-10**, meaning that the model is only trained on ModelNet-10 and directly evaluated on **both** ShapeNet-10 and ScanNet-10; 2) **ShapeNet-10→ModelNet-10/ScanNet-10**; 3) **ScanNet-10→ModelNet-10/ShapeNet-10**. Experimental results demonstrate the effectiveness of SUG framework in learning generalizable features of 3D point clouds, and it can also significantly boost the DG ability for many selected baseline models.

The main contributions of this paper can be summarized as follows:

1) From a new perspective of one-to-many 3D DG, we explore the possibilities of adapting a model from its original source domain to many unseen domains, and study how to leverage the feature's multi-modal information residing in a single dataset.

2) We propose a SUG to tackle the one-to-many 3D DG problem. The SUG consists of a designed MSA method to learn the domain-agnostic and discriminative features during the source-domain training phase, and a SDA strategy to calculate the sample-level inter-domain distance and balance the adaptation degree of different sub-domains with different inter-domain distances.

## 2 RELATED WORKS

### 2.1 2D IMAGE-BASED DOMAIN ADAPTATION AND GENERALIZATION

Recent Domain Adaptation (DA) works can be roughly categorized into two types: 1) Adversarial learning-based methods (Ganin & Lempitsky, 2014; Tzeng et al., 2017; Long et al., 2018b; Kang

et al., 2020) that focus on leveraging a domain label discriminator to reduce the inter-domain discrepancy; 2) Moment matching-based methods (Long et al., 2018a; 2015; Sun & Saenko, 2016) that refer to aligning the first-order or second-order moments of feature distribution. But under the Domain Generalization (DG) setting where the data distribution of the target domain is unavailable, the above DA methods cannot be directly applied to address the DG problem.

For this reason, some researchers (Shankar et al., 2018; Piratla et al., 2020; Chen et al., 2021) start to explore how to adapt the pre-trained model from its source domain to out-of-distribution domain only using source data. For example, some works (Zhu et al., 2022; Zhang et al., 2017) try to boost the model generalization ability using mix-up domains, which generates novel data distribution from the mixtures of multi-domains. Besides, self-supervised learning (SSL) (Wang et al., 2020; Carlucci et al., 2019) also is applied to DG problem to enhance transferable features by leveraging the designed pretext tasks. Although these DG methods have been extensively studied in 2D image tasks, the research on DG problem in 3D point cloud scenarios still remains under-explored, which motivates us to investigate the zero-shot generalization ability of 3D point cloud models.

## 2.2 3D Point Cloud Classification

The existing 3D point cloud classification methods can be divided into: 1) Projection-based and 2) Point-based methods. The projection-based methods first covert irregular points into structured representations, such as multi-view images (Su et al., 2015; Yu et al., 2018), voxels (Riegler et al., 2017), and spherical (Rao et al., 2019). And then, a 2D or 3D neural network is utilized to extract dense features of the structured representations. In contrast, point-based methods (Qi et al., 2017a;b; Wang et al., 2019) directly learn features from the irregular point clouds. This kind of methods can effectively explore the point-wise relations using the designed network such as PointNet (Qi et al., 2017a), which is the first work that directly takes original point clouds as the input and achieves permutation invariance with a symmetric module. Further, considering that point clouds have a variable density at different areas, PointNet++ (Qi et al., 2017b) learns 3D features from multiple semantic levels according to the set abstraction. However, these data-driven point cloud models still face substantial recognition accuracy drop when they are deployed to an unknown domain or dataset.

## 2.3 Domain Adaptation for 3D Point Cloud Classification.

To investigate how to equip a 3D point cloud model with good domain generalization capability, we have reviewed existing domain adaptation-based (Qin et al., 2019; Luo et al., 2021; Shen et al., 2022; Achituve et al., 2021; Yang et al., 2021) or transfer learning-based 3D point cloud works (Ye et al., 2022), and find that they mainly focus on DA study and fail to generalize to **unseen target domains**. These DA-based works can be mainly categorized into two classes: 1) Self-supervised adaptation and 2) Domain-level feature alignment.

For self-supervised adaptation methods (Luo et al., 2021; Shen et al., 2022; Achituve et al., 2021; Yang et al., 2021), they try to design a pretext task to address the common geometric deformations caused by the variances in scanning point clouds. For example, by deforming a region shape of points and reconstructing the original regions of the shape, DefRec (Achituve et al., 2021) can achieve a good domain adaptation result under different domain shift scenarios. Recently, a geometry-aware DA method (Shen et al., 2022) is proposed, which employs the underlying geometric information from points. The domain-level feature alignment method (Qin et al., 2019) focuses on designing an adversarial network to model the discriminative local structures for alignment cross-domain features. Specifically, PointDAN (Qin et al., 2019) proposes a Self-Adaptive (SA) node learning with a node-level attention to present geometric shape information for points.

Overall, when performing the cross-domain adaptation, the above works need to collect extensive target samples in advance for supporting the adaptation process, which is infeasible for many real applications where the target domain is inaccessible or even unknown before deploying the pre-trained model. Thus, it is indispensable that data-driven models trained on a single dataset can handle domain shifts to a certain extent.

## 3 The Proposed Method

The overall SUG framework is illustrated in Fig. 1. For easy understanding, we first give the problem definition of Domain Generalization (DG) for 3D point cloud classification, and then introduce SUG framework including Multi-grained Sub-domain Alignment (MSA) and Sample-level Domain-aware Attention (SDA) modules. Finally, the overall loss function and DG strategy are described.

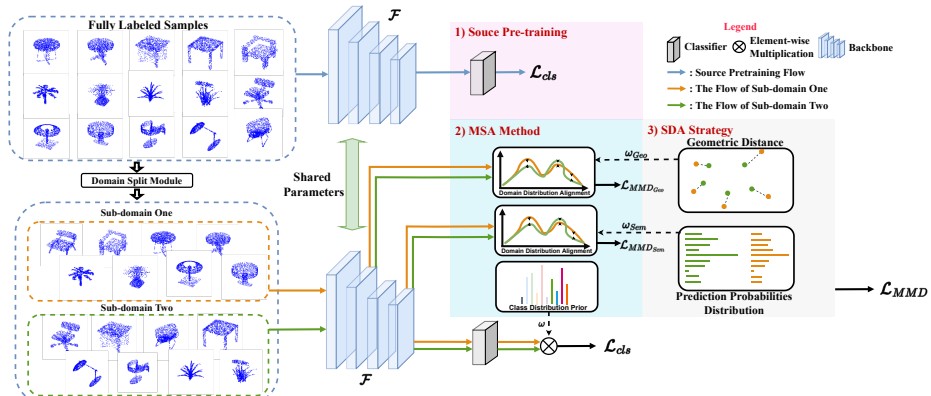

Figure 1: SUG framework, consisting of MSA and SDA to tackle the one-to-many DG.

## 3.1 PRELIMINARIES

**Problem Definition.** Suppose that a domain is defined by a joint distribution $P_{XY}$ on $\mathcal{X} \times \mathcal{Y}$, where $\mathcal{X}$ and $\mathcal{Y}$ stand for the input image and label space, respectively. In the scope of **DG**, $K$ source domains $\mathcal{S} = \left\{ S_k = \left\{ \left( \mathbf{x}^{(k)}, y^{(k)} \right) \right\} \right\}_{k=1}^{K}$ are available for the training process, where each distinct domain is associated with one distribution $P_{XY}^{k}$. And the goal of DG is to obtain a model $f : \mathcal{X} \rightarrow \mathcal{Y}$, trained on the source domain(s), which would have minimized prediction errors on the unseen target domain(s).

Point cloud data is a set of unordered 3D points $\mathbf{x} = \{p_i \mid i = 1, \dots, n\}$, where each point $p_i$ is normally represented by its 3D coordinate $(x, y, z)$ and $n$ is the number of sampling points of one 3D object. We use $(\mathbf{x}, y)$ to denote one training sample pair, and $y$ is its label.

**Single-dataset DG.** In the 3D point-based single-dataset DG setting, the training model *can only access* **one labeled dataset** $\mathcal{S}$, and is required to be evaluated on $M$ unseen target datasets $\mathcal{T}$ (usually $M > 1$). The corresponding joint distribution could be described with $\mathcal{T} = \left\{ T_m = \left\{ \left( \mathbf{x}^{(m)}, y^{(m)} \right) \right\} \right\}_{m=1}^{M}$. Also, $P_{XY}^m \neq P_{XY}^{(k)}, \forall k \in \{1, \dots, K\}, \forall m \in \{1, \dots, M\}$. In our problem set, $\mathcal{Y}_S$ and $\mathcal{Y}_T$ share the same label space. The goal of 3D DG is to improve the performance of source-trained model $f$ on the unseen target domain(s) with the following objectives:

$$min \; \mathbb{E}_{(\mathbf{x},y) \in \mathcal{T}} \; \epsilon(f(\mathbf{x}), y), \tag{1}$$

where $\epsilon$ is the cross-entropy error in our classification task, which can be further defined as:

$$\mathbb{E}_{\mathcal{T}}[-\log p(\hat{y} = c \mid \mathbf{x})], \tag{2}$$

with the prediction that can be obtained with:

$$p(\hat{y} = c \mid x) = \text{softmax}\left( \mathcal{C}_\theta \left( \mathcal{F}_\phi(\mathbf{x}) \right) \right),$$

where $\mathbf{x}$ is the input point cloud instance, $\hat{y}$ is the predicted label. The $\mathcal{F}$ is the embedding network parameterized by $\phi$, and $\mathcal{C}$ is the classifier parameterized by $\theta$.

## 3.2 SUG: A SINGLE-DATASET UNIFIED GENERALIZATION FRAMEWORK

To overcome the two challenges discussed in Sec. 1, we introduce a SUG framework consisting of two novel plug-and-play modules, e.g. **Multi-grained Sub-domain Alignment (MSA)** and **Sample-level Domain-aware Attention (SDA)**, which can be inserted into existing 3D backbones to learn more domain-agnostic representations, to be elaborated in Sec. 3.2.1 and 3.2.2, respectively.

First, the single source dataset is fed into *a designed split module* to get multiple sub-domains of the original source-dataset based on pre-defined heuristics. Then, the embedding network $\mathcal{F}$ takes all the splitted sub-domains as the network input, and converts the point cloud instance $\mathbf{x}$ into multi-level feature vectors $f_l = \mathcal{F}_{\phi,l}(\mathbf{x})$ and $f_h = \mathcal{F}_{\phi,h}(f_l)$, where $f \in \mathbb{R}^{1 \times d}$ and $f_l$ and $f_h$ denote the learned low-level and high-level representations. To handle feature discrepancies from different sub-domains, the MSA module is applied to align the multi-grained features, both at low- and high-level, which can constrain the network to focus on the domain-agnostic representations. Meanwhile, the SDA module is used to selectively enhance the alignment constraints rising from the easy-to-transfer samples to ensure an even adaptation across different sub-domains.

### 3.2.1 Multi-grained Sub-domain Alignment (MSA)

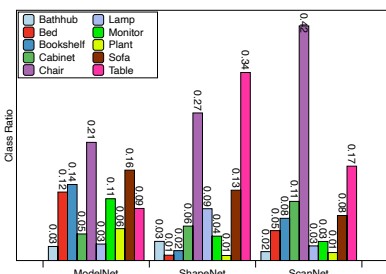

Figure 2: Class distribution shifting across datasets in PointDAN.

**Class distribution alignment.** The 3D point clouds have been deployed in plenty of application scenarios where the objects' distribution shifts significantly, resulting in different distribution patterns residing in different objects, as shown in Fig. 2. To handle such a cross-dataset class-imbalance issue, we incorporate the class-wise sample weighting $\omega$ with the original classification loss (refer to Eq. 2), and the complete weighted classification loss can be written as follow:

$$\mathcal{L}_{cls}(\mathcal{B}) = -\sum_{\mathbf{x} \in \mathcal{B}} \alpha(y) L(\theta; \mathbf{x}), \quad (3)$$

where $\mathcal{B}$ denotes a data batch. The weighting vector $\alpha$ could be set following different heuristics, like FocalLoss (Lin et al., 2017) and DLSA (Xu et al., 2022), etc. Here, we follow the definition in DLSA (Xu et al., 2022), where samples are weighted by:

$$\alpha(i) = \frac{m_i^{-q}}{\sum_j m_j^{-q}}. \quad (4)$$

where $m_i$ is the number of training samples of the class $i$, and $q$ is a positive number controlling the weights distribution. The optimization objective of previous methods such as FocalLoss (Lin et al., 2017) and DLSA (Xu et al., 2022), is to tackle the class imbalance problem within a single dataset, while the optimization function of our method is to tackle the **cross-dataset** class-wise imbalance issue, which is illustrated in Fig. 2. Note that different 3D datasets present an inconsistent class-distribution, which motivates us to use Eq. 4 to learn a uniform and even class-distribution by re-weighting class-distribution for each dataset. Such a way is beneficial to learn more generalizable representations that can avoid to overfit to the class-distribution of the soucre dataset.

**Geometric shifting alignment.** Due to the objects' geometric variances in different scenarios and inconsistent data acquisition procedures, the objects from the same class across different datasets present diverse geometric appearances, as illustrated in Fig. 3(a) across different rows. Meanwhile, the objects' geometric appearance varies greatly with a certain class or a single dataset, which offers the potential that we could use the geometric variances within a single dataset to effectively simulate the ones between different datasets.

To be more specific, we take the low-level feature vector $f_l$ from the shallow layer of the embedding module $\mathcal{F}$, and minimize the Maximum Mean Discrepancy (MMD)(Borgwardt et al., 2006) (Long et al., 2013) loss to align the geometric features from different sub-domains as follows:

$$L_{MMD_{Geo}} = \frac{1}{n_s n_s} \sum_{i,j=1}^{n_s} \kappa\left(f_{l_i}^s, f_{l_j}^s\right) + \frac{1}{n_s n_t} \sum_{i,j=1}^{n_s, n_t} \kappa\left(f_{l_i}^s, f_{l_j}^t\right) + \frac{1}{n_t n_t} \sum_{i,j=1}^{n_t} \kappa\left(f_{l_i}^t, f_{l_j}^t\right), \quad (5)$$

where $\kappa$ is the kernel function, and its superscript $t$ and $s$ denote two different sub-domains sampled from a single dataset. We use the Radial Basis Function (RBF) kernel in our SUG, which is consistent with previous work Qin et al. (2019).

**Semantic variance alignment.** After the high-level features $f_l$ from $\mathcal{F}$ are obtained, the semantic variance alignment is applied to minimize the semantic-level discrepancy between features across different sub-domains before feeding into the classifier. The intuition of the semantic alignment rises from the observation that there exist samples from different classes that could have the similar geometric appearances. As illustrated in Fig. 3(b), the class Table and Cabinet resemble some samples in Chair class as they are all four-legged items. And by conducting semantic variance alignment, the model will learn less single-domain geometric bias yet discriminative representations. The semantic alignment constraints $L_{MMD_{Sem}}$ can be easily calculated by employing the $f_{h_j}^t$ and $f_{h_j}^s$ as the input in Eq. 5.

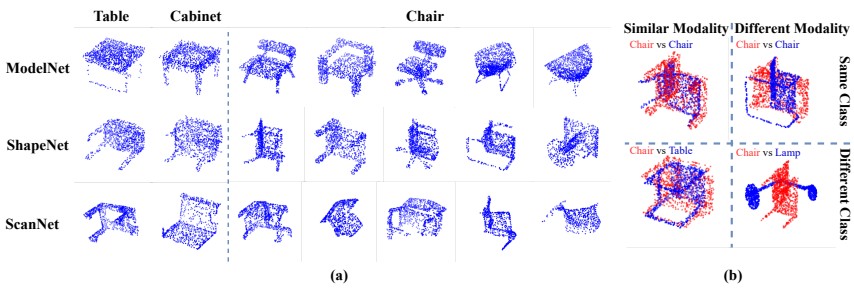

Figure 3: Illustration of distinct characteristics of data in 3D datasets. (a) Geometric and semantic-level domain variances within and between datasets. (b) Geometric similarity comparisons within and between classes.

### 3.2.2 SAMPLE-LEVEL DOMAIN-AWARE ATTENTION (SDA)

The aforementioned MSA module guides the model to learn more domain-agnostic representations. However, the features inside one mini-batch from different sub-domains do not contribute equally to the sub-domain alignment process, since they could contain distinct feature distributions. Ignoring such diversity and imposing equal importance for different samples would result in the hard-to-transfer samples deteriorating the generalization procedure. Meanwhile, the designed domain split module in SUG framework inevitably introduces randomness to different sub-domains with different domain variances, which could also hurt the model generalization performance. Towards safer transfer, we propose the SDA module to enhance the alignment constraints from easy-to-transfer samples. To be more specific, we add sample-level weights $\omega$ to the alignment constraints, inverse proportional to the domain distance $\mathbf{d}$, expressed as:

$$L_{MMD_{weighted}} = \omega * L_{MMD} = \frac{1}{\mathbf{d}} * L_{MMD}, \tag{6}$$

where $\mathbf{d}$ could be realized by using either Eq. 7 or Eq. 8. As for the **geometric shifting alignment**, we use the 3D reconstruction metric as the distance function. In our implementation, Chamfer Distance (CD) is used, which can be formulated as follows:

$$d_{CD}(\mathbf{X}, \mathbf{Y}) = \sum_{x \in \mathbf{X}} \min_{y \in \mathbf{Y}} ||x - y||_2^2 + \sum_{y \in \mathbf{Y}} \min_{x \in \mathbf{X}} ||x - y||_2^2, \tag{7}$$

where $\mathbf{X}$ and $\mathbf{Y}$ are two point cloud instances. The geometric weights $d_{CD}$ focus on the geometric consistency explicit, as shown in the first column of Fig. 3(a), where the samples with geometric similarity have relative small CD distance even they could come from different classes. While for the samples with distinct geometric appearances, the CD distance is higher and the corresponding MMD constraints would be relaxed.

For the **semantic variance alignment**, we adopt the Jensen–Shannon (JS) divergence as our metric. And for symmetric usage, the JS-distance $d_{JS}(\mathbf{X}, \mathbf{Y})$ is defined as:

$$d_{JS}(\mathbf{X}, \mathbf{Y}) = \frac{1}{2} D_{\mathrm{KL}}(\mathbf{X}\|\mathbf{Y}) + \frac{1}{2} D_{\mathrm{KL}}(\mathbf{Y}\|\mathbf{X}), \tag{8}$$

where $D_{\mathrm{KL}}(\mathbf{X}\|\mathbf{Y})$ is the discrete format of KL divergence, represented as:

$$D_{\mathrm{KL}}(X\|Y) = \sum_{c \in \mathcal{C}} X(c) \log\left(\frac{X(c)}{Y(c)}\right), \tag{9}$$

where $X(c)$ and $Y(c)$ are the probability of predicting a sample belonging to the class $c$. In contrast to the geometric weighting, $d_{KL}$ emphasizes more semantic consistency and tends on conducting the alignments among samples belonging to the same class.

### 3.3 OVERALL OBJECTIVES AND DOMAIN GENERALIZATION STRATEGY

**Overall Objectives.** With the MMD constraints introduced in Sec. 3.2.1 and the corresponding weights stated in Sec. 3.2.2, the complete MMD loss could be defined as

$$L_{MMD} = \omega_{Geo} * L_{MMD_{Geo}} + \omega_{Sem} * L_{MMD_{Sem}}. \tag{10}$$

The overall training loss consists of the classification loss as described in Eq. 3 and the MMD loss in Eq. 10, which can be written as follows:

$$L = L_{cls} + \lambda L_{MMD}, \qquad (11)$$

where $\lambda$ is the weighting factor to balance the classification task and the alignment process.

**Domain Generalization Strategy.** We train our model in an end-to-end manner, and the training procedure consists of two steps as follows.

**Step 1:** Firstly, the model is trained using classification loss as defined in Eq. 3, which can ensure that the model learns discriminative features for the subsequent domain transfer. **Step 2:** Secondly, to learn a robust representation that can be generalized to different target datasets, we train the baseline model with the complete loss $L$ as defined in Eq. 11, aiming to constraint the learned representations to be domain-agnostic and discriminative.

## 4 EXPERIMENTS

### 4.1 DATASETS AND IMPLEMENTATION DETAILS

**Datasets.** In order to conduct the experimental evaluation for domain adaptation setting, Point-DAN (Qin et al., 2019) extracts point cloud samples of 10 shared classes from ModelNet40 (Vishwanath et al., 2009), ShapeNet (Chang et al., 2015), and ScanNet (Dai et al., 2017). We follow the work (Qin et al., 2019) and select the same datasets to verify the effectiveness of the proposed method. **ModelNet-10** ($M$) contains a total of 4183 training samples and 856 test samples of 10 classes, which are collected using a 3D CAD model. **ShapeNet-10** ($S$) has 17378 frames for training and 2492 frames for testing, and these frames are produced using a 3D CAD model. **ScanNet-10** ($S^*$) includes a total of 7879 samples that are re-scanned from real-world indoor scene.

**Implementation Details.** For our SUG framework, we employ the PointNet (Qi et al., 2017a) and DGCNN (Wang et al., 2019) as the feature embedding network while the classifier $\mathcal{C}_\theta$ is constructed with a Multi-Layer Perceptron (MLP) using a three-layer fully-connected network. The sample weighting control $q$ in Eq. 4 and the hyper-parameters of $\lambda$ are set to be 0.2 and 0.5, respectively, and the comparison results are shown in Appendix. During the training phase, we use the common naive data augmentations as described in the work (Qin et al., 2019). The Adam optimizer (Kingma & Ba, 2014) is utilized using an initial learning rate of 0.001 and 0.0001, weight decay of 0.00005 and 0.0001 for DGCNN and PointNet backbones, respectively. When testing the generalization performance, to make a fair comparison with the works Zou et al. (2021); Shen et al. (2022), we align each object along x and y axes for the DGCNN backbone and no alignment procedure is applied for experiments on PointNet backbone. During the DG adaptation process, we mainly judge whether the model adaptation state reaches optimal by the designed cross-domain MMD loss. When the change of the MMD loss tends to be stable and has less fluctuation, the adaptation process ends. For all our experiments, we report mean value over the three runs. And we use $\mathcal{F}$'s third layer and $\mathcal{C}_\theta$'s second layer as the low- and high-level features, respectively.

### 4.2 HOW TO SPLIT: DOMAIN SPLIT MODULE DESCRIPTIONS

In this section, we describe the prior-knowledge based domain split modules and employ the corresponding modules to conduct experiments, as shown in Table 1. Note that our splitting procedure is conducted via a class-wise manner within a source dataset to ensure that each sub-domain contains all categories of the dataset. Please refer to Appendix for more discussions about the hand-designed domain split modules.

**Random Splitting.** We conduct the random sampling and split a single source dataset into different sub-domains with the same sample size, where domain characteristic of each sub-domain is identical with that of the original one.

**Geometric Splitting.** In our practice, we randomly select one sample as the anchor sample of a certain class, then compute the Iterative Closest Point (ICP) registration score between other samples of the current class and the selected anchor sample. After getting all registration scores of all samples, the current class is clustered into $K$ sub-domains according to the calculated score.

**Entropy Splitting.** We quantify the uncertainty of classifier's predictions with the entropy criterion $H(\mathbf{g}) = -\sum_{c=1}^{C} X_c \log X_c$, as defined in Eq. 9. Note that the classifier used here is pre-trained model on the source domain. For the domain split module, the single dataset is clustered into $K$ sub-domains based on the entropy scores of all samples.

**Feature Clustering Splitting.** We infer the whole dataset with the pre-trained model on source dataset, and save the feature maps before feeding them to the classifier. After that, we use Principal

Table 1: Results on different domain split methods under the **one-to-many** Domain Generalization (DG) setting. **Avg** denotes the mean adaptation accuracy across all target domains. The results of Random Splitting are averaged over three runs, and we report mean value over the three runs.

| Domain Split Method | Setting | Backbone | $M$ as Source Domain | | | $S$ as Source Domain | | | $S^*$ as Source Domain | | |
| --- | --- | --- | --- | --- | --- | --- | --- | --- | --- | --- | --- |
| | | | $M \to S$ | $M \to S^*$ | **Avg.** | $S \to M$ | $S \to S^*$ | **Avg.** | $S^* \to M$ | $S^* \to S$ | **Avg.** |
| w/o Adapt | Source-Only | PointNet | 42.5 | 22.3 | 32.4 | 39.9 | 23.5 | 31.7 | 34.2 | 46.9 | 40.6 |
| PointDAN (NeurIPS'19) | UDA | PointNet | 64.2 | 33.0 | 48.6 | 47.6 | 33.9 | 40.8 | 49.1 | 64.1 | 56.6 |
| Random Splitting (Sec. 4.2) | DG | PointNet | 54.5 | 36.3 | 45.4 | 37.8 | 31.7 | 34.8 | 45.0 | 53.0 | **49.0** |
| Geometric Splitting (Sec. 4.2) | | | 57.4 | 41.7 | **49.6** | 30.3 | 31.6 | 31.0 | 38.3 | 44.2 | 41.3 |
| Entropy Splitting (Sec. 4.2) | | | 55.4 | 42.5 | 49.0 | 36.5 | 27.7 | 32.1 | 41.7 | 49.9 | 45.8 |
| Feature Clustering Splitting (Sec. 4.2) | | | 60.4 | 36.1 | 48.3 | 45.4 | 31.7 | **38.6** | 37.6 | 45.6 | 41.6 |
| Random Splitting (Sec. 4.2) | DG | DGCNN | 80.8 | 53.2 | **67.0** | 69.4 | 49.5 | 59.5 | 61.4 | 57.6 | 59.5 |
| Geometric Splitting (Sec. 4.2) | | | 79.3 | 49.9 | 64.6 | 56.7 | 53.2 | 55.0 | 40.5 | 64.4 | 52.5 |
| Entropy Splitting (Sec. 4.2) | | | 73.6 | 49.3 | 61.5 | 72.8 | 50.3 | **61.6** | 42.9 | 60.9 | 51.9 |
| Feature Clustering Splitting (Sec. 4.2) | | | 77.8 | 52.9 | 65.4 | 71.0 | 47.6 | 59.3 | 63.0 | 59.3 | **61.2** |

Table 2: Results on PointDA-10 under the **one-to-many** Domain Generalization (DG) setting. **Note that** our SUG can be *simultaneously* generalized to multiple target domains without accessing any target samples. In contrast, UDA methods can only be adapted to a single target domain. For example, GAST model performs an adaptation from the domain $M$ to another domain $S$, but the adapted model cannot perform well in a new domain $S^*$.

| Method | Setting | Backbone | $M$ as Source Domain | | | $S$ as Source Domain | | | $S^*$ as Source Domain | | |
| --- | --- | --- | --- | --- | --- | --- | --- | --- | --- | --- | --- |
| | | | $M \to S$ | $M \to S^*$ | **Avg.** | $S \to M$ | $S \to S^*$ | **Avg.** | $S^* \to M$ | $S^* \to S$ | **Avg.** |
| w/o Adapt | Source-Only | PointNet | 42.5 | 22.3 | 32.4 | 39.9 | 23.5 | 31.7 | 34.2 | 46.9 | 40.6 |
| | | DGCNN | 83.3 | 43.8 | 63.6 | 75.5 | 42.5 | 59.0 | 63.8 | 64.2 | 64.0 |
| PointDAN (NeurIPS'19) | UDA | PointNet | 64.2 | 33.0 | 48.6 | 47.6 | 33.9 | 40.8 | 49.1 | 64.1 | 56.6 |
| GAST (ICCV'21) | UDA | DGCNN | 84.8 | 59.8 | 72.3 | 80.8 | 56.7 | 68.8 | 81.1 | 74.9 | 78.0 |
| SLT (CVPR'22) | UDA | DGCNN | 86.2 | 58.6 | 72.4 | 81.4 | 56.9 | 69.2 | 81.5 | 74.4 | 77.9 |
| **our SUG** | DG | PointNet | 64.3 | 40.7 | 52.5 | 44.0 | 36.2 | 40.1 | 44.5 | 54.7 | 49.6 |
| | | DGCNN | 82.8 | 57.2 | 70.0 | 74.8 | 52.2 | 63.5 | 74.1 | 64.6 | 69.4 |

Component Analysis (PCA) with t-SNE (Van der Maaten & Hinton, 2008) to get the dimensional reduced representations and apply $K$-means to get $K$ sub-domain clusters.

### 4.3 HOW TO ALIGN: DOMAIN-AGNOSTIC FEATURE LEARNING

**DG Baseline Implementation.** In this part, we study how to use the off-the-shelf UDA technique to achieve unseen domain generalization. First, we use the domain split module to generate different sub-domain data. Then, when a source dataset is clustered into $K$ sub-domains, 3D UDA methods such as PointDAN (Qin et al., 2019) can be used to perform a sub-domain adaptation within a single dataset. In our baseline practice, we directly use the implementation from PointDAN (Qin et al., 2019) without any further modification, in order to align the feature gaps between different splitted sub-domains. It can be seen from Table 1 that, by leveraging the above domain split modules to split a single dataset into different sub-domains, the off-the-shelf UDA method (Qin et al., 2019) can **simultaneously** boost the model generalization ability for **multiple unseen datasets**. It also can be concluded that, multi-modal distribution exists within a single-source dataset. As a result, a hand-designed domain split method coupled with the off-the-shelf UDA baseline can significantly boost unseen domain generalization. Besides, we also observe that the classification accuracy of the model in the target domain is related to the selected network structure. This is intuitive since different network structures have different model capacities that can learn features with the different sensitivities to the source-to-target feature variations.

**SUG Implementation.** Although a naive UDA baseline coupled with our designed domain split modules can enhance the model's zero-shot recognition ability, it is still important for one-to-many adaptation to exploit multi-modal feature variations across different sub-domains, and further learn as many domain variances as possible. We conduct the experiments using the designed MSA and SDA, and the results are shown in Table 2 and Random Splitting is applied to obtain sub-domains.

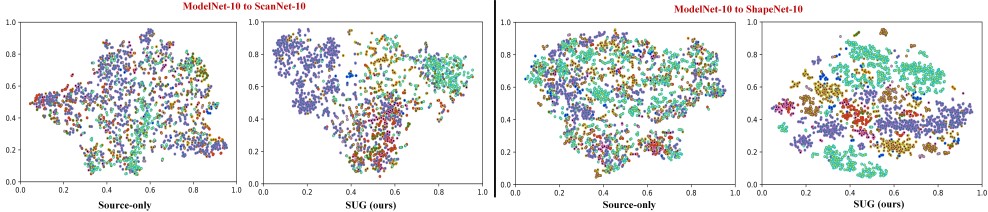

Figure 4: tSNE results (PointNet). Different colors denote different classes.

Table 3: Results on down-sampling the whole source dataset using different methods.

| Down-sampling Methods | Data Diversity | $M \to S$ | $M \to S^*$ | Avg. | Source Accuracy | Train Sample Size |
|---|---|---|---|---|---|---|
| **Split & Select** (Feature Clustering Splitting) | Low | 48.9 | 33.6 | 41.3 | 81.2 | 1015 |
| **Split & Select** (Geometric Splitting) | Low | 53.7 | 45.0 | 49.4 | 80.1 | 975 |
| **Random Sampling** | High | 55.4 | 45.2 | 50.3 | 88.8 | 1044 |

Table 4: Ablation studies of class-wise classification accuracy, where the model is trained on ModelNet-10 and directly evaluated on ScanNet-10 (M→S*). PointNet is used as backbone.

| Methods | CD Align. | GS Align. | SV Align. | SDA | Bathtub | Bed | Bookshelf | Cabinet | Chair | Lamp | Monitor | Plant | Sofa | Table | Avg. |
|---|---|---|---|---|---|---|---|---|---|---|---|---|---|---|---|
| Supervised | | | | | 88.9 | 88.6 | 47.8 | 88.0 | 96.6 | 90.9 | 93.7 | 57.1 | 92.7 | 91.1 | 83.5 |
| w/o Adapt | | | | | 59.4 | 1.0 | 18.4 | 7.4 | 55.7 | 43.5 | 84.8 | 60.0 | 3.4 | 39.7 | 37.3 |
| PointDAN | | | | | 84.7 | 1.6 | 19.0 | 1.3 | 81.9 | 63.3 | 90.5 | 82.3 | 2.2 | 82.9 | 51.0 |
| | ✓ | | ✓ | ✓ | 68.1 | 2.6 | 20 | 0.0 | 49.0 | 53.9 | 95.3 | 86.4 | 0.3 | 79.5 | 45.5 |
| | ✓ | ✓ | | ✓ | 64.5 | 5.6 | 17.0 | 0.7 | 75.2 | 61.4 | 90.5 | 78.6 | 0.2 | 88.4 | 48.2 |
| **our SUG** | ✓ | ✓ | ✓ | ✓ | 64.1 | 0.0 | 17.8 | 1.43 | 75.9 | 55.7 | 92.0 | 90.0 | 0.0 | 85.0 | 48.2 |
| | ✓ | ✓ | ✓ | | 65.9 | 5.0 | 34.0 | 4.0 | 74.1 | 58.4 | 91.7 | 77.3 | 0.0 | 86.7 | 49.7 |
| | | | | ✓ | 80.9 | 0.0 | 19.1 | 0.0 | 73.3 | 63.8 | 93.7 | 72.5 | 0.0 | 80.9 | 48.4 |
| | ✓ | ✓ | ✓ | ✓ | 76.9 | 2.0 | 25.0 | 2.0 | 81.5 | 57.6 | 89.7 | 88.2 | 0.4 | 85.0 | **50.8** |

First, our results show that the state-of-the-art 3D-based UDA methods (Zou et al., 2021; Shen et al., 2022) cannot work well under the one-to-many generalization scenario. For example, GAST (Zou et al., 2021) can obtain a relatively high result (84.8%) under $M \to S$ setting, but the adapted model has a serious accuracy drop (only 40.1%) under another target domain $M \to S^*$ for our experiments. This is mainly because these methods often try to perform the explicit cross-domain alignment between the source domain and a specific target domain, which is hard to ensure that the adapted model has an even generalization toward different domains. In contrast, our SUG achieves higher one-to-many zero-shot generalization results for different target domains (*e.g.* 82.8% for $S^*$ and 57.2% for $S$).

To validate that SUG can be generalized to different point cloud backbones, we also conduct the DG experiments on many backbones (DGCNN, Point Transformer Zhao et al. (2021), and KP-Conv Thomas et al. (2019)). It can be seen from Table 2 that SUG yields consistent accuracy gains.

**SUG Limitation.** Our SUG framework assumes that the source domain dataset presents multi-modal feature distributions which can be implicitly exploited to model the feature distribution differences residing in the multi-modal distributions. In 3D scenario, our assumption holds since the 3D point cloud samples for each class often have diverse appearances, geometric shapes, and *etc*, as shown in Fig. 3. Here, we further discuss the limitation cases of our SUG from: **the diversity of source domain distribution gradually decreases**.

To this end, we first split the given single dataset into $M$ sub-domains, and then select one of the sub-domains from the splitting results (1 out of 4 splits) as the training set, which is described in Sec. 4.2 and denoted as **Split & Select**. For comparison, we also randomly sample from the complete set of the given single dataset, which is denoted as **Random Sampling**. The biggest difference between the above down-sampling methods is that a single split (sub-domain) has much less data diversity and domain variances inside than the randomly sampled one that has a similar distribution status as the original dataset. It can be seen from Table 3 that, when the data distribution within the sampled sub-domain becomes more undiversified, the zero-shot generalization ability of model from a source domain to multi-target domains will drop.

### 4.4 FURTHER ANALYSES

**Ablation Studies.** In Table 4, we conduct the ablation studies from the following two aspects: 1) MSA method that consists of Class Distribution (CD Align.), Geometric Shifting (GS Align.), and Semantic Variance (SV Align.) alignments; 2) SDA strategy. First, MSA learns the domain-agnostic features from various granularities including class-level, geometry-level, and semantic-level. We observe that each newly-added alignment constraint can bring accuracy gains. Besides, we also conduct experiments of removing the SDA to investigate the effectiveness of the designed SDA. The results shown in Table 4 demonstrate that, by enhancing some easy-to-adapt instances to keep an even adaptation, SDA significantly boosts the generalization accuracy from 48.4% to 50.8%.

**tSNE Results.** We visualize features from source-only model and our SUG under in Fig. 4. The visualizations show that features learned by SUG can improve the model discriminability of different classes' features from unseen domains. For more visualization results, please see Fig. 7-8 in Sec. A.

## 5 CONCLUSION

We have proposed a SUG framework to tackle the one-to-many Domain Generalization (DG) problem in 3D scenario. SUG consists of a MSA method to exploit the data diversity residing in a given source dataset and further learn domain-agnostic and discriminative representations, a SDA strategy to selectively increase the domain adaptation degree for easy-to-adapt instances. Experiments are conducted on public benchmarks to show the effectiveness of SUG in tackling the 3D DG problem.

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

# A  APPENDIX

## A.1  DISCUSSION OF THE HAND-DESIGNED DOMAIN SPLIT MODULES

In Table 1 of the main text, we conduct the extensive experiments to show the Domain Generalization (DG) results for different domain split modules. However, through extensive experiments, we observe that these DG results achieved by the domain split modules are unstable for different cross-domain settings. Here, we give **two main reasons** for such instability as follows.

1) The distribution shift patterns across datasets are quite different. ModelNet and ShapeNet are both CAD-generated datasets. As a result, they contain similar geometric characteristics, and at least both of them are without occlusions and follow a similar appearance. In this way, using "Feature Clustering" to emphasize the semantic discrepancy and alignment would bring more gains (As reported in Table 1 for M → S and S → M experiments). In contrast, ScanNet is obtained from the real world and originally designed for segmentation tasks. In other words, it is quite different both in semantic and geometric views, as shown in Fig. 3. In such a situation, emphasizing solely geometric or semantic discrepancy is not optimal while random splitting is a strong baseline to conduct the alignment operation, and this phenomenon is consistent with our experiment results in S* → M, S* → S, and S → S* cross-domain settings.

2) The split results achieved by the domain splitting strategy are quite imbalance along the sample size. Take "Entropy Clustering" as an example, since the pre-trained model (on source-dataset) with source domain-related distribution characteristics is used, the model will be quite confident on predicting the source-domain samples, resulting in a quite imbalance clustering result. For example, the PointNet backbone on ScanNet dataset will get 3504 samples *vs.* 2606 samples splitting results for each sub-domain. But it will get worse on the easier dataset like ModelNet, where 2542 samples *vs.* 1641 samples splitting results for each sub-domain. Such imbalance is harmful for model training since that will bring the bias from the source dataset characteristic to the training procedure.

Table 5: The number of parameters of the backbone networks employed by SUG.

| Network | Parameters |
|---|---|
| PointNet Qi et al. (2017a) | 3.5M |
| DGCNN Wang et al. (2019) | 1.8M |
| PointTransformer Zhao et al. (2021) | 9.6M |
| KPConv Thomas et al. (2019) | 5.3M |

Table 6: Results on PointDA-10 under the **one-to-many** Domain Generalization (DG) setting with additional backbones *e.g.* KPConv and Point Transformer.

| Method | Setting | Backbone | $M$ as Source Domain | | |
|---|---|---|---|---|---|
| | | | $M \rightarrow S$ | $M \rightarrow S^*$ | **Avg.** |
| w/o Adapt | Source-Only | KPConv | 81.76 | 46.06 | 63.91 |
| | | Point Transformer | 84.11 | 54.83 | 69.47 |
| **our SUG** | DG | KPConv | 81.08 | 47.67 | **64.38** |
| | | Point Transformer | 83.36 | 58.35 | **70.86** |

## A.2  COMBINATION WITH POINT TRANSFORMER AND KPCONV

To further verify the superiority of our SUG in boosting more baseline models, we select two state-of-the-art 3D point-cloud backbone networks, *e.g.,* Point Transformers Zhao et al. (2021) and KPConv Thomas et al. (2019) to conduct the one-to-many DG study. The corresponding experimental results are shown in Table 6.

According to the above experimental results shown in Table 6, we summarize the following two main empirical findings.

1) By coupling with our method, the Point Transformer Zhao et al. (2021) can achieve a better one-to-many DG classification performance gain, such 1.39 for M → S, M → S* settings. But it should be pointed out that the accuracy gain of Point Transformer is relatively slight compared with that

of the DGCNN backbone. This is mainly due to the fact that the transformer-based methods could learn much discriminative features during the model training phase, which is consistent with the observations in Allen et al. (2019). But Point Transformer Zhao et al. (2021) also takes much more time in model training and hyper-parameter tuning (As reported in Table 5).

2) The DG classification performance gain on KPConv Thomas et al. (2019) is quite minor, which is mainly due to that, the dataset-related parameter settings like query radius, are sensitive to different target domains. Besides, we observe that, during the inference process where the points selected by the kernel of KPConv in ModelNet are generally more than 100 points (the first layer), but less than 80 points selection could happen if we did not change that parameters when used for ScanNet. The cross-domain feature alignment process brings more negative effects towards source-similar ModelNet than positive gains towards source-dissimilar ScanNet, which results in a lower average classification accuracy across different datasets.

### A.3 QUALITATIVE ANALYSES

**More tSNE results between source-only model and our SUG.** In the main text, we have shown the tSNE visualization results of high-level features learned by the source-only model and our SUG, respectively. In this part, we give more tSNE visualization results for more cross-domain settings such as the adaptation from ShapeNet-10 to ModelNet-10, ShapeNet-10 to ScanNet-10, *etc*. As illustrated in Fig. 8 to 7, these visualization results demonstrate that the features from an unseen target domain (*e.g.*, ModelNet-10) have a distinct feature discrimination for different classes, further verifying that the learned features are domain-agnostic and discriminative for unseen domains.

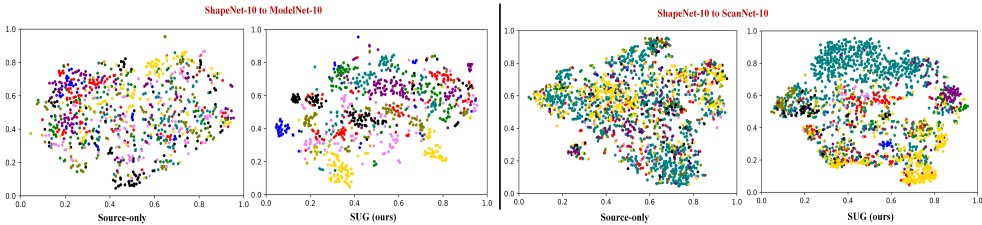

Figure 5: tSNE results of ModelNet-10 and ScanNet-10 datasets, where the model is trained on ShapeNet-10 dataset and different colors denote different classes.

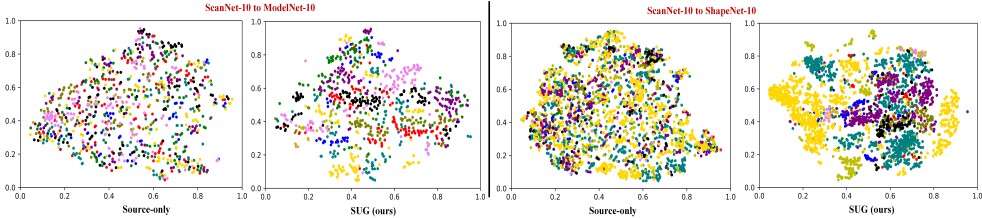

Figure 6: tSNE results of ModelNet-10 and ShapeNet-10 datasets, where the model is trained on ScanNet-10 dataset and different colors denote different classes.

**More tSNE results of domain split modules.** In this part, we split the training dataset (source domain) into two sub-domains. And then we use the baseline model and the proposed SUG to train on those two sub-domains, respectively. After the training process ends, we visualize and compare the extracted features of the baseline model and the proposed SUG by t-SNE, respectively. The visualization results are shown in Fig. 8.

**More tSNE results of sub-domains characteristics using random splitting module.** Moreover, in order to validate the consistency of the distribution from sub-domains characteristics with the Random Splitting module, we split a single source dataset into different sub-domains using the random sampling strategy. Then we use the pre-trained model to extract features from each sub-domain and tSNE is applied to compare the features. The visualization results are shown in Fig. 9.

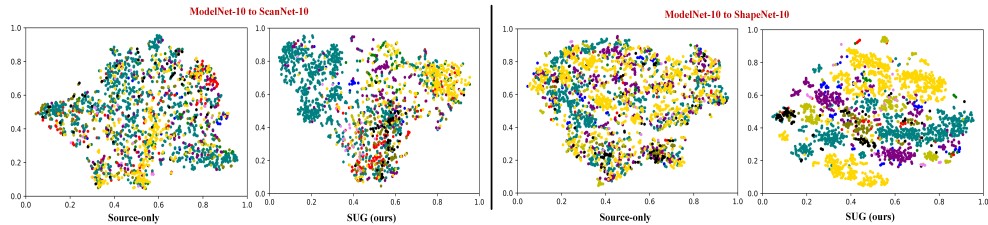

Figure 7: tSNE results of ScanNet-10 and ShapeNet-10 datasets, where the model is trained on ModelNet-10 dataset and different colors denote different classes.

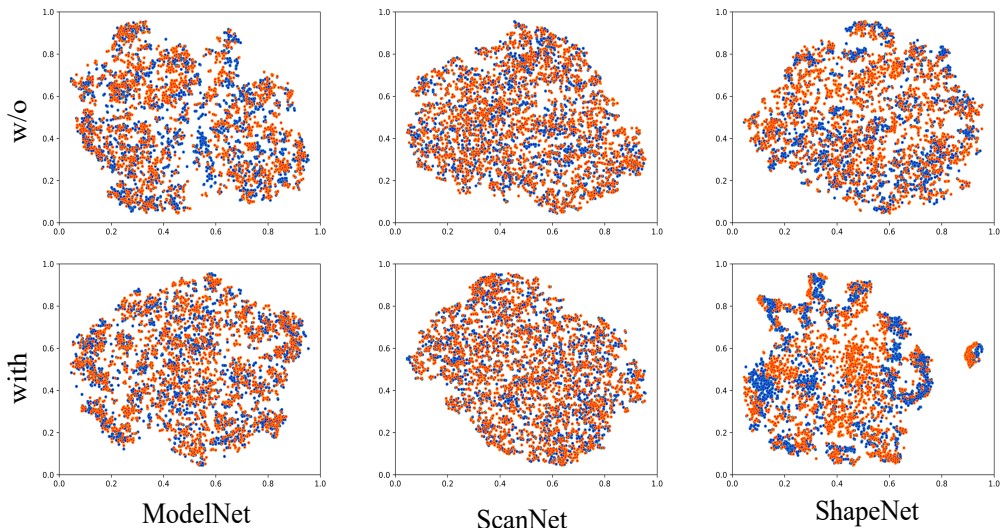

Figure 8: tSNE results of sub-domains without and with alignment module. The first and second rows show the learned features without or with the feature alignment process, respectively. Different colors denote features from different sub-domains.

## A.4    DISCUSSION ON THE ALIGNMENT CONSTRAINTS.

**Discussion on the scope of MMD constraints.** Generally speaking, the alignment should be conducted between different modalities. And as shown in Fig. 3(b), similar modalities exist across classes while different modalities exist within a single class. As a result, we are expected to fully exploit multiple modality information from both intra- and inter-classes, and thus do not perform a hard class-wise MMD-based alignment. Specifically, to avoid losing the label information, we first turn the class label into a scaled one-hot vector and then concatenate it with the feature maps before conducting the MMD alignment, which is termed as the "Soft-MMD".

Besides, we have implemented different MMD-based alignment methods by changing the class-label information constraint, such as "Hard-MMD" which means that only samples from the same class are aligned, and "Max-Hard MMD" which means that we first re-order the samples from different domains to let them have most class overlapping, and then conduct the Hard-MMD. Through our experiments, we found that Soft-MMD outperforms other MMD-based alignment designs, as shown in Table 7.

**Comparison between Contrastive Loss and MMD loss.** Contrastive Loss (CL) is also known for its capability of constraining learned features. We conducted experiments to compare the performance between CL and MMD loss used in SUG framework. Actually, for the implementation of Contrastive Loss, we directly use the Pytorch Implementation Pytorch (2022), which is a variation of the Hadsell et al. (2006) with cosine distance.

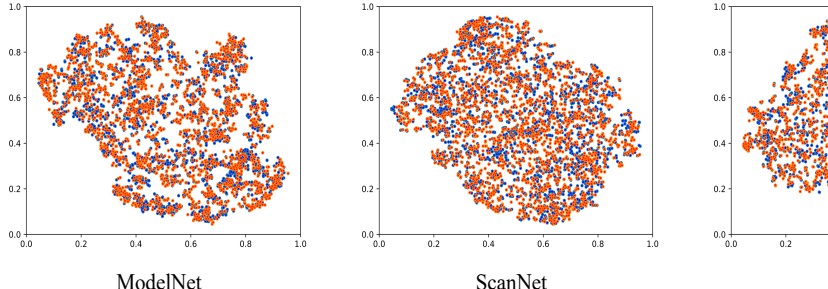

| ModelNet | ScanNet | ShapeNet |

Figure 9: tSNE results of different sub-domains divided by Random Splitting module without using domain alignment. Different colors denote features from different sub-domains.

Table 7: Average results of unseen domains $S$ and $S^*$ among different MMD-based alignment methods, and we employ the $M$ as the source domain.

| Alignment | Avg. Results |
| --- | --- |
| Soft-MMD | 52.5 |
| Hard-MMD | 51.8 |
| Max-Hard-MMD | 52.3 |

Specifically, for the CL, we have to explicitly define the positive and negative pairs, which is quite complex in our setting. Positive pairs are the samples with similar geometric appearances for geometric alignments, regardless of whether they are from the same class. In contrast, the positives are always from the same class for semantic alignments. For simplicity, we directly take the geometric features as negative pairs when they come from different classes under the CL constraint.

Experimentally, we use ModelNet as the source domain and evaluate on ShapeNet and ModelNet. We report the average results on these two datasets. Note that we have yet to tune the parameter for CL loss much.

Based on the above experimental results, we summarize the following empirical findings.

1) As we can see from Table 8, when we replace the MMD loss with CL loss for semantic-level alignment, the final results are still competitive since both CL and MMD can make learned features to be domain-invariant. However, the results for CL loss for geometric-level alignment are much worse. The main reason behind those accuracy differences is that CL focuses on capturing the high-level feature variances while it tends to ignore some low-level information for describing domain shift.

2) Based on the experiments in Table 8, we are delighted that the SUG has the potential to be a unified framework where the sub-domain alignment module could be replaced using other recently-proposed alignment loss function such Contrastive Loss.

Table 8: Average results of unseen domains $S$ and $S^*$ using the Contrastive Loss (CL) and MMD alignment designs, and we employ the $M$ as the source domain. Geo and Sem stand for geometric and semantic alignment.

| Alignment | Geo-MMD | Geo-CL | Sem-MMD | Sem-CL | Avg. Results |
| --- | --- | --- | --- | --- | --- |
| 1-MMD Original | X | | X | | 0.5245 |
| 2-CL | | X | | X | 0.4598 |
| 3-MIX | X | | | X | 0.5234 |
| 4-Mix | | X | X | | 0.4725 |

## A.5 DISCUSSION ON HYPER-PARAMETERS IN SUG.

The experiments in this part are conducted employing PointNet as the backbone and ModelNet as the source domain. We report the average prediction results on ShapeNet and ScanNet.

**Weight $\lambda$.** The $\lambda$ in Eq. 11 achieves a trade-off between the classification task and the alignment process. We conduct the ablation studies to study the sensitivity of the $\lambda$ value setting on our SUG performance. The corresponding results are shown in Table. 9.

**Batch Size $\mathcal{B}$.** We also conduct the experiments of changing the batch-size value, where we keep other settings as default. The results are shown in Table. 10.

Table 9: Average results of unseen domains $S$ and $S^*$ using different $\lambda$ values in Eq. 11, and we employ the $M$ as the source domain.

| $\lambda$ | Avg. Results |
|------|------|
| 0.25 | 44.7 |
| 0.50 | 52.5 |
| 0.75 | 51.2 |
| 1.0 | 47.8 |
| 2.0 | 49.1 |
| 3.0 | 48.5 |
| 4.0 | 45.9 |
| 5.0 | 44.7 |

Table 10: Average results of unseen domains $S$ and $S^*$ trained with different batch sizes, and we employ the $M$ as the source domain.

| Batch Size | Avg. Results |
|------|------|
| 16 | 51.35 |
| 32 | 52.86 |
| 64-Default | 52.45 |
| 128 | 50.45 |
| 256 | 50.52 |
| 512 | 47.51 |

Table 11: Average results of unseen domains $S$ and $S^*$ trained with different layer selection settings, and we employ the $M$ as the source domain.

| Embedding Module Layer | Avg. Result | Classification Module Layer | Avg. Result |
|------|------|------|------|
| Layer-1 | 48.8 | Layer-1 | 49.9 |
| Layer-2 | 49.7 | Layer-2(Default) | 52.5 |
| Layer-3(Default) | 52.5 | Layer-3 | 48.2 |
| Layer-4 | 49.4 | - | - |
| Layer-5 | 48.2 | - | - |

According to the above experimental results shown in Table 10, we find that our method can achieve good generalization ability across different batch-size settings. For the batch-size setting with small value, the mini-batch data could not contain enough information related to the domain distribution. As a result, the SUG could not learn the domain-invariant features well. In contrast, it can be observed that the degradation of generalization's ability when we continuously enlarge the batch size, which is mainly due to that the large-batch training procedure tends to converge to sharp minimizers Keskar et al. (2016).

**Layer selection for low-level and high-level features.** In the default SUG setting, we use $\mathcal{F}$'s third layer and $\mathcal{C}_\theta$'s second layer as the low and high-level features, respectively. To further explore how the layer selection for features would affect the SUG performance, we change the selection choices of the layers. Specifically, in order to validate the choice for geometric features, we use the features from $\{1, 2, 3, 4, 5\}$-layer of the embedding module as the geometric features while keeping the second layer of the classification module as default. For semantic features experiments, we used the features from $\{1, 2, 3\}$-layer of the classification module while keeping the third layer of the embedding module as default.

Based on the above experimental results, we summarize the following empirical findings.

1) For Embedding Module Layer Selection: The features from too shallow layers (**e.g.**, Layer-1) contain much less information and would be sensitive to noise. In contrast, if we choose the features from too deeper layers, the geometric and fine-grained information would be overtaken by the deep semantic information. At the same time, when we choose that deeper features, the geometric alignment would be much similar to semantic alignment and thus lose its discriminability.

2) For Classification Module Layer Selection: The features from the shallow layer (**e.g.**, Layer-1) are similar to the geometric ones and would lose semantic alignment ability. At the same time, the last layer's features are too high-level and lose a lot of semantic information.

