# OpenReview forum: "SUG: Single-dataset Unified Generalization for 3D Point Cloud Classification"
_ICLR.cc/2023/Conference — Submitted to ICLR 2023_

### Official Review · Reviewer_AoJ5 · 2022-10-23

**Confidence:** 4
**Correctness:** 2
**Technical Novelty And Significance:** 2
**Empirical Novelty And Significance:** 3
**Recommendation:** 5

**Clarity, Quality, Novelty And Reproducibility:**

Clarity: clear, and well-written

Quality: many evaluations are missing

Novelty: the addressed problem is interesting

Reproducibility: not sure, no code is available in the submission.

**Strength And Weaknesses:**

=== Strength ===

1. The studied problem is interesting and important, since domain generalization in 2D image has been studied extensively while in 3D point cloud still under-explored.

2. The method is introduced clearly and is easy to follow.

=== Weaknesses ===

1. The sub-domain splitting strategy. In Tab.1, we can find that the performance of different splitting strategies is not stable, as the performance varies from different datasets. A reasonable explanation should be given. In addition, the features of the sub-domains with/without alignment should be compared. For example, extract a validation set from the training set, then train two networks (baseline, the proposed), then visualize the features and make a comparison. I am wondering whether the subdomain splitting makes sense or not.

2. Following the above comment, in fact, the performance of the splitting strategy is also related to the network structure, which makes me confused about the effectiveness of the strategy.

3. This paper only evaluates two very old networks, PointNet and DGCNN. As many SOTA networks are developed, more evaluations are expected, such as KPConv, and Point Transformer.

4. In Page 5, the paper claims that the motivation behind weighted classification loss is different from previous methods. I am so confused. In fact, in this paper, this loss still addresses the class imbalance problem, and there is no difference from other methods.

5. other concerns: how to select the best model for evaluation? how about the variances of the results over three runs?

**Summary Of The Paper:**

This paper studies the single domain generalization problem for 3D point cloud classification, which aims to generalize the model trained on a single source domain to an unknown target domain. The method splits the training domain into two subdomains, and models the alignment between the two sub-domains, expecting the model to learn domain-generalized features. The evaluation is conducted on ModelNet-10, shapeNet10, and ScanNet-10 by considering two networks, PointNet and DGCNN.

**Summary Of The Review:**

The addressed problem is interesting and important but the experiments are not adequate.


=================== post-rebuttal =====================
In the response, the authors have made efforts to conduct additional experiments to address my questions. However, I still have a main conern: different split strategies have different impacts for performance on different datasets and networks.

---

> ### Author Response · Authors · 2022-11-14
> **Response and action to R3 - Part 1**
>
> **R3.1: The performance of different splitting strategies is not stable, as the performance varies from different datasets. A reasonable explanation should be given.**
>
> **A3.1**：Thanks the Reviewer for this valuable comment. We would like to emphasize that we employ the four **widely-used** domain split methods mentioned in the main text and mainly show the existence of multi-modal distribution within a single-source dataset by employing these domain split modules to divide the entire source dataset into different sub-domains. The four **widely-used domain split methods are not our main technical contribution**. By leveraging the simple domain-splitting method coupled with the domain alignment method (such as MMD loss), our main contribution is to indicate that the multi-modal distribution exists within a single-source dataset, as illustrated in Fig. 3 in the main text.
>
> It is meaningful that we found the existence of multi-modal distribution within a single-source dataset, which motivates the researchers to fully exploit the within-dataset domain-feature variations to learn more generalizable representations rather than solely using the previous classification loss to overfit to the data distribution of the source dataset.
>
> Further, according to the Reviewer's comments, we summarized the following two reasons why these widely-used sub-domain splitting methods are **unstable** across different datasets:
>
> - The distribution shift patterns across datasets are quite different. ModelNet and ShapeNet are both CAD-generated datasets. As a result, they contain similar geometric characteristics, and at least both of them are without occlusions and follow a similar appearance. In this way, using “Feature Clustering” to emphasize the semantic discrepancy and alignment would bring more gains (As reported in Table 1 for M→S and S→ M experiments). In contrast, ScanNet is obtained from the real world and originally designed for segmentation tasks. In other words, it is quite different both in semantic and geometric views, as shown in Fig. 3. In such a situation, emphasizing solely geometric or semantic discrepancy is not optimal, while random splitting is a strong baseline to conduct the alignment, and this phenomenon is consistent with our experiment results in S* → M, S* → S, and S → S*.
> - The split results achieved by the domain splitting strategy are quite imbalance along the sample size. Take “Entropy Clustering” as an example, since the pre-trained model (on source-dataset) with source domain-related distribution characteristics is used, the model will be quite confident on predicting the source-domain samples, resulting in a quite imbalance clustering result. For example, the PointNet backbone on ScanNet dataset will get 3504 samples vs 2606 samples splitting results for each sub-domain. But it will get worse on the easier dataset like ModelNet, where 2542 samples vs 1641 samples splitting results  for each sub-domain. Such imbalance is harmful for model training since that will bring the bias from the source dataset characteristic to the training procedure.
>
> Based on the above analyses, we chose to use the Random Splitting method for our SUG to achieve a better one-to-many DG result. According to this comment from the R3, we have supplemented the detailed explanations and experimental results in the Appendix of the revised version due to the page limitation.
>
> **R3.2: In fact, the performance of also related to the network structure, which makes me confused about the effectiveness of the strategy.**
>
> **A3.2:** Thanks for this valuable comment. As stated in Sec.4.2 of the main text, the domain splitting methods are based on the source-dataset pre-trained models, which brings the implicit dependencies over the model structure and network parameters. Consistent with previous UDA works[Ref-1] or DG works[Ref-2], the classification accuracy of the model in the target domain is indeed related to the selected network structure. This is intuitive since different network structures have different model capacities that can learn features with different sensitivities to the source-to-target feature variations. According to R3's comment, we have supplemented the detailed explanations in the experimental evaluation in Sec. 4.1 of the revised version.
>
> [Ref-1] Zou, Longkun, et al. "Geometry-aware self-training for unsupervised domain adaptation on object point clouds." *Proceedings of the IEEE/CVF International Conference on Computer Vision*. 2021.
>
> [Ref-2]Zhang, Chongzhi, et al. "Delving deep into the generalization of vision transformers under distribution shifts." *Proceedings of the IEEE/CVF Conference on Computer Vision and Pattern Recognition*. 2022.

---

> > ### Author Response · Authors · 2022-11-14
> > **Response and action to R3 - Part 2**
> >
> > **R3.1: In addition, the features of the sub-domains with/without alignment should be compared. For example, extract a validation set from the training set, then train two networks (baseline, the proposed). Then visualize the features and make a comparison.**
> >
> > **A3.1:** We thank the Reviewer very for the valuable comment. Accordingly, we split the training dataset into two sub-domains. And then, we use the baseline model and our proposed SUG to train on those two sub-domains. After the training procedure, we visualized and compared those two sub-domain features. The corresponding visualization results are shown in Fig. 8 in the Appendix of the revised version.
> >
> > **R3.3: This paper only evaluates two very old networks, PointNet and DGCNN. As many SOTA networks are developed, more evaluations are expected, such as KPConv and Point Transformers.**
> >
> > **A3.3:** Thank the Reviewer very much for these insightful comments. Actually, **all** recently-public  Unsupervised Domain Adaptation (UDA)-based 3D point cloud works use the PointNet or DGCNN backbone to conduct the experiments. As a result, we follow previous 3D point cloud-based works [Ref-3], [Ref-4], [Ref-5], to employ the typical 3D backbone (such as PointNet and DGCNN) to ensure a fair comparison.
> >
> > Nevertheless, we accept the Reviewer's suggestion and include more state-of-the-art networks, such as PointTransformers and KPConv, to conduct the DG study. The corresponding experimental results are shown as follows.
> >
> > **Table 1: DG Results on PointDA-10 with KPConv and Point Transformer.**
> > | Method | Setting | Backbone | ShapeNet | ScanNet | Average |
> > | --- | --- | --- | --- | --- | --- |
> > | w/o Adapt | Source-only | KPConv | 81.76 | 46.06 | 63.91 |
> > |  |  | Point Transformer | 84.11 | 54.83 | 69.47 |
> > | our SUG | DG | KPConv | 81.08 | 47.67 | 64.38 |
> > |  |  | Point Transformer | 83.36 | 58.35 | 70.86 |
> >
> > Based on the above experimental results, we summarize the following **two** main empirical findings.
> >
> > - By coupling with our method, the Point Transformer [Ref-6] can achieve better one-to-many DG results, such as 1.39 for M→S, M→S* settings. However, it should be pointed out that the accuracy gain of the Point Transformer is relatively slight compared with that of the DGCNN backbone. This is mainly because the transformer-based methods could learn many discriminative features during the model training phase, which is consistent with the observations in  [Ref-7]. However, it also takes much more time in model training and hyper-parameter tuning (As reported in Table 2).
> >
> > **Table 2: Parameters Number across different backbones.**
> > | Network | Parameters |
> > | --- | --- |
> > | PointNet | 3.5M |
> > | DGCNN | 1.8M |
> > | PointTransformer | 9.6M |
> > | KPConv | 5.3M |
> >
> > - The generalization's gain on KPConv [Ref-8] is relatively minor, mainly because the dataset-related parameter settings, like query radius, are sensitive to different target domains. Besides, we observe that, during the inference process where the points selected by the kernel of KPConv in ModelNet are generally more than 100 points (the first layer), but less than 80 points selection could happen if we did not change the parameters when used for ScanNet. The cross-domain feature alignment process brings more negative effects toward source-similar ModelNet than positive gains toward source-dissimilar ScanNet, which results in a lower average classification accuracy across different datasets.
> >
> > According to the Reviewer's comment, we have supplemented the experimental results of Point Transformer and KPConv in Sec. 4.3 of the revised version, and the experimental analyses in the Appendix of the revised version, due to the page limitation.
> >
> > [Ref-3]Qin, Can, et al. "Pointdan: A multi-scale 3d domain adaption network for point cloud representation." *Advances in Neural Information Processing Systems* 32 (2019).
> >
> > [Ref-4]Zou, Longkun, et al. "Geometry-aware self-training for unsupervised domain adaptation on object point clouds." *Proceedings of the IEEE/CVF International Conference on Computer Vision*. 2021.
> >
> > [Ref-5]Wu, Pingyu, Wei Zhai, and Yang Cao. "Background activation suppression for weakly supervised object localization." *2022 IEEE/CVF Conference on Computer Vision and Pattern Recognition (CVPR)*. IEEE, 2022.
> >
> > [Ref-6]Zhao, Hengshuang, et al. "Point transformer." *Proceedings of the IEEE/CVF International Conference on Computer Vision*. 2021.
> >
> > [Ref-7]Zhang, Chongzhi, et al. "Delving deep into the generalization of vision transformers under distribution shifts." *Proceedings of the IEEE/CVF Conference on Computer Vision and Pattern Recognition*. 2022.
> >
> > [Ref-8]Thomas, Hugues, et al. "Kpconv: Flexible and deformable convolution for point clouds." *Proceedings of the IEEE/CVF international conference on computer vision*. 2019.APA

---

> > > ### Author Response · Authors · 2022-11-14
> > > **Response and action to R3 - Part 3**
> > >
> > > **R3.4: In Page 5, the paper claims that the motivation behind weighted classification loss is different from previous methods. I am so confused. In fact, this loss still address the class imbalance problem, and there is no difference from other methods.**
> > >
> > > **A3.4:** We sincerely thank the Reviewer for pointing it out. We would like to emphasize that the optimization objective of previous methods, such as FocalLoss[Ref-9], is to tackle the class imbalance problem **within a single dataset,** while the optimization function of our method is to tackle the **cross-dataset** class-wise imbalance issue, which is illustrated in Fig. 2. Note that different 3D datasets present an inconsistent class-distribution, which motivates us to use Eq.4 to learn a uniform and even class-distribution by re-weighting class-distribution for each dataset. Such a way is beneficial to learn more generalizable representations that can avoid overfitting the source dataset's class distribution.
> > >
> > > According to this comment raised by the Reviewer, we have added a detailed comparison of the weighted classification loss between the proposed method and the existing methods. Please refer to Page 5 in the revised version for more details.
> > >
> > > [Ref-9] Lin, Tsung-Yi, et al. "Focal loss for dense object detection." *Proceedings of the IEEE international conference on computer vision*. 2017.
> > >
> > > **R3.5: Other concerns: how to select the best model for evaluation? how about the results over three runs?**
> > >
> > > **A3.5:** During the DG adaptation process, we mainly judge whether the model adaptation state reaches optimal by the designed cross-domain MMD loss. When the change of the MMD loss tends to be stable and has less fluctuation, the adaptation process ends.  In fact, some recent works [Ref-10] have utilized the same strategy.
> > > According to R3's comment, we have supplemented the experimental implementation descriptions of the model evaluation method in Sec. 4.1 of the revised version.
> > >
> > > [Ref-10] Self-training and adversarial background regularization for unsupervised domain adaptive one-stage object detection. ICCV-2019

---

### Official Review · Reviewer_Yd9F · 2022-10-24

**Confidence:** 3
**Correctness:** 3
**Technical Novelty And Significance:** 3
**Empirical Novelty And Significance:** 3
**Recommendation:** 8

**Clarity, Quality, Novelty And Reproducibility:**

The paper is easy to follow. It provides a reasonable method and comprehensive experiments with sufficient analysis.

**Strength And Weaknesses:**

Strengths:
1.	This work is technically sound and shows promising results.
2.	The design of the proposed SUG framework fits well in the one-to-many DG task.
3.	The extensive experiments on 3 well-established datasets verify the efficiency of the proposed approach.

Weakness:
1.	The differenences, characteristics and comparisons of different datasets has not been fully discussed.
2.	Lack of introduction to DGCNN and motivation for choosing to use DGCNN as the backbone network.


**Summary Of The Paper:**

This work proposes a Single-dataset Unified Generalization (SUG) framework for the challenging one-to-many domain generalization task in 3D point clouds. A Multi-grained Sub-domain Alignment (MSA) is proposed for spliting the single dataset into multiple sub-domains and then constraining the learned representations to be domain-agnostic and discriminative. A Sample-level Domain-aware Attention (SDA) is further proposed for adapting the uneven domain to an even inter-domain. Extensive experiments on several benchmarks illustrate the effectiveness of SUG.

**Summary Of The Review:**

Overall this work is well-motivated and fits well in the one-to-many domain generalization task in 3D point clouds.

---

> ### Author Response · Authors · 2022-11-16
> **Response and action to R2**
>
> **Q2.1** **The differences, characteristics and comparisons of different datasets has not been fully discussed.**
>
> **A2.1** We sincerely thank the Reviewer for this valuable suggestion. And we add more details regarding to the datasets used in the experiments as follows:
>
> - The ModelNet-10 contains 4183 training and 856 testing samples of 10 classes collected using 3D CAD models. Each point cloud sample is sampled from the surface following [Ref-1]. Moreover, each sample contains 2048 points.
> - The ShapeNet-10 has 17378 frames for training and 2492 frames for testing, and each point cloud frame has 2048 points. These points are sampled uniformly on the surface of ShapeNet objects. Due to the larger scale, the ShapeNet contains more structure variance and heterogeneous information.
> - The ScanNet-10 has 6110 training samples and 1769 testing samples. Each point cloud frame is obtained by collecting 2048 points from partially visible object point clouds contained in the annotated bounding box. As ScanNet contains scanned and reconstructed real-world indoor scenes, the point clouds usually lose parts due to the occlusion with the contextual objects. Moreover, the point clouds generally with noise due to the realistic sensor noises.
>
> [Ref-1] Qi, Charles R., et al. "Pointnet: Deep learning on point sets for 3d classification and segmentation." *Proceedings of the IEEE conference on computer vision and pattern recognition*
> . 2017.
>
> **Q2.2 Lack of introduction to DGCNN and motivation for choosing to use DGCNN as the backbone network.**
>
> **A2.2** We sincerely thank the Reviewer for pointing this out. And we add more details to the DGCNN method and the motivation for choosing it as follows.
>
> - **All** recently-public Unsupervised Domain Adaptation (UDA)-based 3D point cloud works use the PointNet or DGCNN backbone to conduct the experiments. As a result, we follow previous 3D point cloud-based works [Ref-2] [Ref-3] [Ref-4] to employ the typical 3D backbone (such as PointNet and DGCNN) to ensure a fair comparison.
> - DGCNN, dubbed EdgeConv, acts on graphs computed in each network layer. One of its most significant characteristics is that it models point clouds as connected graphs, which are dynamically built using k-nearest neighbors in the latent space. Moreover, it learns per-point features with message passing.
>
> [Ref-2]Zou, Longkun, et al. "Geometry-aware self-training for unsupervised domain adaptation on object point clouds." *Proceedings of the IEEE/CVF International Conference on Computer Vision*. 2021.
>
> [Ref-3]Achituve, Idan, Haggai Maron, and Gal Chechik. "Self-supervised learning for domain adaptation on point clouds." *Proceedings of the IEEE/CVF winter conference on applications of computer vision*. 2021.
>
> [Ref-4]Shen, Yuefan, et al. "Domain Adaptation on Point Clouds via Geometry-Aware Implicits." *Proceedings of the IEEE/CVF Conference on Computer Vision and Pattern Recognition*
> . 2022.

---

> > ### Comment · Reviewer_Yd9F · 2022-12-14
> > **Thanks for the response.**
> >
> >  I think my concerns have been well addressed. Thus I'd like to maintain my positive stance.

---

### Official Review · Reviewer_i2R8 · 2022-10-25

**Confidence:** 5
**Correctness:** 3
**Technical Novelty And Significance:** 3
**Empirical Novelty And Significance:** 3
**Recommendation:** 5

**Clarity, Quality, Novelty And Reproducibility:**

A. Clarity: needs improvement (see my detailed comments in the summary section)

B. Quality: acceptable but not strong enough (see my detailed comments in the summary section)

C. Novelty: the problem setting is new, but the proposed solution has limited practicality (see my detailed comments in the summary section)

D. Reproducibility: since many technical details are missing, it is hard to reproduce the work (see my detailed comments in the summary section).

**Strength And Weaknesses:**

A. Strength
- The studied problem is important.
- The proposed method and its components are well-motivated.

B. Weaknesses
- Presentation needs improvement (many technical details are missing; notation is not used consistently)
- Experiments are lacking in some aspects.
- The proposed method does not outperform state-of-the-art (SLT, CVPR22)
- The performance is quite low compared with supervised learning approach, limiting the practicality of the proposed method.

**Summary Of The Paper:**

The paper proposes a method for domain generalization for 3D point clouds-based deep learning (i.e., pre-training a 3D model on a source domain to well generalize on a new and unseen domain without accessing any data points from the new domain). The proposed method includes three main components: domain splitting (to split a source domain into sub-domains), multi-grained sub-domain alignment (to align features at different levels from sub-domains), and sample-level domain-aware attention (to weight samples based on their distances to their domain). The proposed method was experimented with two point cloud-based backbones (PointNet, DGCNN) and evaluated on benchmark datasets (ModelNet, ShapeNet, and ScanNet), and also compared with existing ones on these datasets.

**Summary Of The Review:**

My detailed comments for the paper are as follows.

A. Presentation:
- Many technical details are missing. For instance, it is not clear how the mini-batches can be created to include data points from different sub-domains. It is also not clear what layers should be considered to extract low-level and high-level features. What is the kernel function used in Eq (5) in experiments? Do i and j in Eq (5) need to represent samples of the same class? Also, Eq (5) assumes that there are two sub-domains, but what if there are more than 2 domains? What are X and Y in Eq (7) and Eq (8), are they from the same class or different classes? If X and Y come from different classes, then d in Eq (6) cannot show the domain distance. What is lambda in "Implementation Details"? What setting (e.g., splitting method) is used for SUG in Table 2 and Table 4? Why the results of SUG in Table 2 do not look like any results of SUG in Table 1?
- Notation is not used consistently, leading to confusion. For instance, in Eq (4), n is referred to as the number of samples in a class but it is not so in Eq (5). Also, do n_s and n_t represent the number of samples in sub-domains s and t? The same notation for omega in Eq (4) and Eq (6), but, I believe, these omega parameters are different in these two equations. In 3.1, K is referred to as sub-domains, while denoted as M in 4.3.

B. Methodology
- For geometric splitting, what if Chamfer distance is used to compare two different point clouds?
- I was wondering what if a contrastive loss is used for the L_MMD_Geo in Eq (5) as contrastive loss is also known for the capability of constraining learnt features. A comparison between the MMD and contrastive loss would be useful to consolidate the proposed solution.

C. Experiments and results
- How many sub-domains are used in experiments and how is the method affected by the number of sub-domains?
- I was wondering how the method performs under different batch sizes, as the method may be affected when the batches do not well reflect the data distributions in sub-domains.
- It is important to experiment the method with different selections of the layers for the low-level and high-level features
- As mentioned in comments for methodology, it would be great to see a comparison between MMD and contrastive loss.
- ScanObjectNN in [a] is a real-world dataset including practical challenges such as occlusions, background intervention, etc., that do not exist in ModelNet, and therefore would be ideal to validate the domain generalization ability of the proposed method. Note that, this dataset also shows that many models, successful on ModelNet, fail on it.
- The proposed method does not outperform state-of-the-art (e.g., SLT)
- The role of SDA is not very clear (given experimental results). And, it may also be affected by batch setting. This should be verified.
- Also in terms of performance, SUG is well below supervised learning approach. On the one hand, I do understand that the studied problem is much more challenging than supervised learning approach. On the other hand, the performance of SUG shows that its current status is not at the applicable level.

D. Others
- It would be clear to explicitly state that d in Eq (6) can be realized using either Eq (7) or Eq (8).
- It is not clear how the domain characteristic of each sub-domain is identical with the original domain from the following claim "We conduct the random sampling and split a single source dataset into different sub-domains with the same sample size, where domain characteristic of each sub-domain is identical with that of the original one". It may be over claimed.

E. Missing reference
- [a] Revisiting Point Cloud Classification: A New Benchmark Dataset and Classification Model on Real-World Data, ICCV19.

---

> ### Author Response · Authors · 2022-11-16
> **Response and action to R1 - Part 1**
>
> **Q1.1: How the mini-batches can be created to include data from different sub-domains.**
>
> Thanks to the Reviewer for this valuable comment. The data from different sub-domains are directly constructed from different **dataloaders**. The total mini-batch size is equal to the addition between the source-domain (sub-domain one) mini-batch size and the target-domain (sub-domain two) mini-batch size. These two sub-domain min-batch sizes are identical in our implementation. If not stated especially, the batch size value is referred to the source/target mini-batch size.
>
> **Q1.1: What layers should be considered to extract low-level and high-level features.**
>
> We sincerely thank the Reviewer for pointing this out. In the SUG paper, we use the embedding module's third layer and the classification module's second layer for low and high-level features, respectively. See details in our Response A1.6.
>
> **Q1.1: What is kernel function used in Eq.(5). Do i and j need to represent samples with the same class**
>
> Thanks to the Reviewer for this valuable comment.  We use the Radial Basis Function (RBF) kernel in our SUG, which is consistent with previous works like [Ref-1].
>
> [Ref-1] Qin, Can, et al. "Pointdan: A multi-scale 3d domain adaption network for point cloud representation." *Advances in Neural Information Processing Systems* 32 (2019).
>
> **Q1.1: What if there are more than 2 sub-domains?**
>
> Thanks to the Reviewer for pointing it out. Since the current alignment is done between two arbitrary sub-domains, we assume that this alignment process could be done iteratively and taken in turn. With such an iterative manner, more sub-domains could be aligned.
>
> According to R1’s comment, we have updated the expression in the revised version regarding the above presentation issues.
>
> **Q1.1: What is lambda in “Implementation Details”?**
>
> Thanks to the Reviewer for this valuable comment. The complete expression of Eq.(11) should be: L = L_{cls} + lambda * L_{MMD}, achieving a trade-off between the classification task and the alignment process. We conduct ablation studies to study the sensitivity of the lambda value setting on our SUG performance. The corresponding results are shown in Table 1. We conduct experiments with PointNet as the backbone and use ModelNet as the source domain.
>
> **Table 1: Average results of unseen domains S and S^{*} using varied lambda values in Eq.(11), and we employ the M as the source domain.**
> | Lambda  | Avg. Results |
> | --- | --- |
> | 0.25 | 44.7 |
> | 0.5 | 52.5 |
> | 0.75 | 51.2 |
> | 1.0 | 47.8 |
> | 2.0 | 49.1 |
> | 3.0 | 48.5 |
> | 4.0 | 45.9 |
> | 5.0 | 44.7 |
>
> According to R1’s comment, we have updated the expression of Eq.(11) in Sec. 3.3 of the revised version, and the experimental analyses in the Appendix of the revised version, due to the page limitation.
>
> **Q1.1: What are X and Y in Eq.(7) and Eq.(8)? Are they from the same class or different classes?**
>
> We sincerely thank the Reviewer for this valuable comment. Generally speaking, the alignment should be conducted between different modalities. And as shown in Fig. 3(b), similar modalities exist across classes, while different modalities exist within a single class. As a result, we are expected to exploit multiple modality information from both intra- and inter-classes fully and thus do not perform a hard class-wise MMD-based alignment. Specifically, to avoid losing the label information, we first turn the class label into a scaled one-hot vector and then concatenate it with the feature maps before conducting the MMD alignment, which is termed the “Soft-MMD.”
>
> Besides, we have implemented different MMD-based alignment methods by changing the class-label information constraints, such as “Hard-MMD,” which means that only samples from the same class are aligned, and “Max-Hard MMD,” which means that we first re-order the samples from different domains to let them have most class overlapping and then conduct the Hard-MMD. Through our experiments, we found that Soft-MMD outperformed others, as shown in Table 2.
>
> **Table 2: Average results of unseen domains S and S^{*} among different MMD-based alignment methods, and we employ the M as the source domain.**
>
> | Alignment | Avg. Results |
> | --- | --- |
> | Soft-MMD | 52.5 |
> | Hard-MMD | 51.8 |
> | Max-Hard-MMD | 52.3 |
>
> According to this comment raised by R1, we have added a detailed comparison of the alignment methods. Please refer to Page 15 in the revised version for more details.

---

> > ### Author Response · Authors · 2022-11-16
> > **Response and action to R1 - Part 2**
> >
> > **Q1.1: What setting (e.g. splitting method) is used for SUG in Table 2 and Table 4? Why the results of SUG in Table 2 do not look like any results of SUG in Table 1?**
> >
> > - Thanks to the Reviewer for pointing it out. In Table 2 and Table 4, we chose to use the Random Splitting method.
> > - The results in Table 1 are not with our proposed SUG but rather with naive alignment. To be more specific, we directly use the open-source UDA framework[Ref-2], where we took the split domains as the source and target domains, respectively, and conducted the domain alignment. We want to use Table 1 to validate the existence of sub-domains inside a single dataset and show the potential to boost the generalization's ability with domain alignment methods.
> >
> > [Ref-2] [https://github.com/canqin001/PointDAN](https://github.com/canqin001/PointDAN)
> >
> > **Q1.1  Notation is not used consistently**.
> >
> > Thanks to the Reviewer for this valuable comment. We have updated our notation for consistency. Specifically, we use m_i to stand for the training samples of class i and use alpha in Eq (3) to represent the sample weights. Moreover, we use K to represent sub-domains throughout the paper. These updates are also in our revisions.
> >
> > **Methodology 2:**
> >
> > **Q1.2: For geometric sampling, what if Chamfer Distance is used to compare two different point clouds?**
> >
> > **A1.2:** Thanks for this valuable comment. According to the Reviewer's comment, we add the Chamfer Distance as the metric to conduct the geometric sampling and train our SUG on the ModelNet dataset. The experimental results are shown as follows.
> >
> > **Table 3: Comparison between ICP Score and Chamfer Distance used for dataset splitting.**
> >
> > | Split Method | Metric | Setting | ShapeNet | ScanNet | Avg. Results |
> > | --- | --- | --- | --- | --- | --- |
> > | Geometric Splitting | ICP Score | DG | 57.4 | 41.7 | 49.6 |
> > | Geometric Splitting | Chamfer Distance | DG | 55.0 | 41.4 | 48.2 |
> >
> > Based on the experimental results in Table 2, we summarize the following empirical findings.
> >
> > - The Chamfer Distance (CD) could also be one comparable metric for the geometric splitting procedure. Moreover, the final DG average performance is slightly worse than the ICP-based methods.
> > - ICP and CD metrics could describe the geometric similarity between two point clouds. However, since ICP is in the iterative manner, where the Rotation Matrix and translation vector are optimized during the registration process, in such a way, the ICP score could take the view differences (while with similar geometric appearances) into consideration. Moreover, that situation (similar appearances, different views) is expected in the point could datasets.
> > - Since ICP is conducted sequentially and iteratively and is hard to be optimized for parallel computing and thus takes much more time than Chamfer Distance, the CD would be a good choice for a large-scale dataset.

---

> > > ### Author Response · Authors · 2022-11-16
> > > **Response and action to R1 - Part 3**
> > >
> > > Methodology 2:
> > >
> > > **Q1.3:** **I was wondering what if contrastive loss is used for the L_MMD_Geo in Equ.5. A comparison between MMD and contrastive loss would be useful to consolidate the proposed solution.**
> > >
> > > **A1.3:**  We sincerely thank the Reviewer for this suggestion.  Contrastive Loss (CL) is also known for its capability of constraining learned features. We conducted experiments to compare the performance between CL and MMD loss used in the SUG framework. Actually, for the implementation of Contrastive Loss, we directly use the Pytorch Implementation [Ref-3], which is a variation of [Ref-4] with cosine distance.
> > >
> > > Specifically, for the CL, we have to explicitly define the positive and negative pairs, which is quite complex in our setting. Positive pairs are the samples with similar geometric appearances for geometric alignments, regardless of whether they are from the same class. In contrast, the positives are always from the same class for semantic alignments. Due to the time limits, we have to simplify our setting where we directly take the geometric features as negative pairs when they come from different classes when we use the contrastive loss for geometric alignment.
> > >
> > > Experimentally, we use ModelNet as the source domain and evaluate on ShapeNet and ModelNet. We report the average results on these two datasets. Note that we have yet to tune the parameter for CL loss much.
> > >
> > > **Table 4: Average results of unseen domains S and S^{*} using the Contrastive Loss (CL) and MMD alignment functions.**
> > >
> > > | Alignment  | Geo-MMD | Geo-CL | Sem-MMD | Sem-CL | Avg. Result |
> > > | --- | --- | --- | --- | --- | --- |
> > > | 1-SUG Original | X |  | X |  | 0.5245 |
> > > | 2-2CL |  | X |  | X | 0.4598 |
> > > | 3-Mix | X |  |  | X | 0.5234 |
> > > | 4-Mix |  | X | X |  |  |
> > >
> > > Based on the above experimental results, we summarize the following empirical findings.
> > >
> > > - As we can see from Table 4, when we replace the MMD loss with CL loss for semantic-level alignment, the final results are still competitive since both CL and MMD can make learned features to be domain-invariant. However, the results for CL loss for geometric-level alignment are much worse. The main reason behind those accuracy differences is that CL focuses on capturing the high-level feature variances while it tends to ignore some low-level information for describing domain shifts.
> > > - Based on the experiments in Table 3, we are delighted that the SUG has the potential to be a unified framework where the sub-domain alignment module could be replaced using other recently-proposed alignment loss functions, such as Contrastive Loss.
> > >
> > > [Ref-3] [https://pytorch.org/docs/stable/generated/torch.nn.CosineEmbeddingLoss.html](https://pytorch.org/docs/stable/generated/torch.nn.CosineEmbeddingLoss.html)
> > >
> > > [Ref-4] Hadsell, Raia, Sumit Chopra, and Yann LeCun. "Dimensionality reduction by learning an invariant mapping." *2006 IEEE Computer Society Conference on Computer Vision and Pattern Recognition (CVPR'06)*. Vol. 2. IEEE, 2006.
> > >
> > > According to R1’s comment, we have supplemented the experimental results of Contrastive Loss in the revised version, and the experimental analyses in the Appendix A.4 of the revised version, due to the page limitation.
> > >
> > > **Q1.4: How many sub-domains are used in experiments, and how is the method affected by the number of sub-domains?**
> > >
> > > **A1.4:** Thanks to the Reviewer for this valuable comment.  Considering that our SUG is developed as one extension and forward-step for previous Unsupervised Domain Adaptation (UDA) frameworks, where generally two domains are aligned in one alignment process, two sub-domains are used in the experiments.

---

> > > > ### Author Response · Authors · 2022-11-16
> > > > **Response and action to R1 - Part 4**
> > > >
> > > > **Q1.5: I was wondering how the method performs under different batch sizes, as the method would be affected when the batches does not well reflect the data distribution in the sub-domains.**
> > > >
> > > > **A1.5:** Thanks to the Reviewer for this inspiring suggestion.  According to the Reviewer's comment, we conducted the experiments on the batch size where we varied the batch sizes and kept others as default. The experiment is conducted with the PointNet backbone and takes the ModelNet as the source domain.
> > > >
> > > > **Table 5: Average results of unseen domains S and S^{*} trained with different batch sizes, and we employ the M as the source domain.**
> > > > | Batch size | Avg. Result |
> > > > | --- | --- |
> > > > | 16 | 51.35 |
> > > > | 32 | 52.86 |
> > > > | 64-default | 52.45 |
> > > > | 128 | 50.45 |
> > > > | 256 | 50.52 |
> > > > | 512 | 47.51 |
> > > >
> > > > According to the above experimental results shown in Table 5, we find that our method can achieve good generalization ability across different batch-size settings. For the batch-size setting with a small value, the mini-batch data could not contain enough information related to the domain distribution. As a result, the SUG could not learn the domain-invariant features well. In contrast, it can be observed that the degradation of generalization's ability when we continuously enlarge the batch size, which is mainly due to the large-batch training procedure tends to converge to sharp minimizers [Ref-5].
> > > >
> > > > [Ref-5] Keskar, Nitish Shirish, et al. "On large-batch training for deep learning: Generalization gap and sharp minima." *arXiv preprint arXiv:1609.04836* (2016).
> > > >
> > > > According to R1’s comment, we have supplemented the experimental results of Contrastive Loss in the revised version, and the experimental analyses in the Appendix A.4 of the revised version, due to the page limitation.
> > > >
> > > > **Q1.6:** **It is important to experiment the method with different selections of the layers of the low-level and high-level features.**
> > > >
> > > > **A1.6:** Thanks to the Reviewer for this valuable suggestion. We have added the corresponding experiments to show the results of aligning different network layers.
> > > >
> > > > In the default SUG setting, we use the third layer of the embedding module F and the classification module's second layer as the low and high-level features, respectively. To further explore how the layer selection for features would affect the SUG performance, we change the selection choices of the layers. Specifically, in order to validate the choice for geometric features, we use the features from {1,2,3,4,5}-layer of the embedding module as the geometric features while keeping the second layer of the classification module as default. For semantic features experiments, we used the features from {1,2,3}-layer of the classification module while keeping the third layer of the embedding module as default. The experiment results are shown as follows.
> > > >
> > > > **Table 6: Average results of unseen domains S and S^{*} trained with different layer selection settings.**
> > > > | Embedding Module Layer | Average  | Classification Module Layer | Average |
> > > > | --- | --- | --- | --- |
> > > > | Layer-1 | 48.8 | Layer-1 | 49.9 |
> > > > | Layer-2 | 49.7 | Layer-2(Default) | 52.5 |
> > > > | Layer-3 (Default) | 52.5 | Layer-3 | 48.2 |
> > > > | Layer-4 | 49.4 |  |  |
> > > > | Layer-5 | 48.2 |  |  |
> > > >
> > > > Based on the above experimental results, we summarize the following empirical findings.
> > > >
> > > > - For Embedding Module Layer Selection: The features from too shallow layers (e.g., Layer-1) contain much less information and would be sensitive to noise. In contrast, if we choose the features from too deeper layers, the geometric and fine-grained information would be overtaken by the deep semantic information. At the same time, when we choose that deeper features, the geometric alignment would be much similar to semantic alignment and thus lose its discriminability.
> > > > - For Classification Module Layer Selection: The features from the shallow layer (e.g., Layer-1) are similar to the geometric ones and would lose semantic alignment ability. At the same time, the last layer's features are too high-level and lose a lot of semantic information.
> > > >
> > > > According to R1’s comment, we have supplemented the experimental results of different layer selections in the revised version, and the experimental analyses in the Appendix A.5 of the revised version, due to the page limitation.
> > > >
> > > > **Q1.7: ScanObjectNN is a real-world dataset including practical challenges such as occlusions, background intervention, etc., that do not exist in ModelNet, and therefore would be ideal to validate the domain generalization ability of the proposed method.**
> > > >
> > > > We thank the Reviewer very much for the valuable comment. The SUG proposed in this paper holds the assumption that the source domain and the target domain have the same label space, e.g., label-space consistency. However, the ScanObjectNN dataset obtains 15 classes, with only six classes aligned with the datasets in Point-DA10, which violates our method's assumption. And we would leave that OOD adaption for future work.

---

> > > > > ### Author Response · Authors · 2022-11-16
> > > > > **Response and action to R1 - Part 5**
> > > > >
> > > > > **Q1.8:** **The proposed method does not out-perform the state-of-the-art (e.g.STL).**
> > > > >
> > > > > We thank the Reviewer very much for the valuable comment. However, we would like to emphasize that STL framework is the work for the UDA setting, while the proposed method SUG is designed for the DG task where we cannot access the target domain data. At the same time, UDA is designed for one-to-one adaption tasks, while DG is designed for one-to-many generalization tasks, and the latter task is much more difficult. To our best knowledge, SUG is the first work towards DG setting in point cloud classification task when it is submitted to ICLR.
> > > > >
> > > > >
> > > > > **Q1.9: Also in terms of performance, SUG is well below supervised learning approach.**
> > > > >
> > > > > We thank the Reviewer very much for the valuable comment. The SUG is designed for Domain Generalization task setting, which is quite challenging since it cannot access the target domain data. And SUG can boost the model's zero-shot generalization ability.
> > > > >
> > > > > **Q1.10: It would be clear to explicitly state that d in Eq.(6) that can be realized using either Eq.(7) or Eq.(8).**
> > > > >
> > > > > We sincerely thank the Reviewer for this valuable suggestion.  We have updated that in our paper to the revised version.
> > > > >
> > > > > **Q1.11: Overclaim over the domain characteristics.**
> > > > >
> > > > > We sincerely thank the Reviewer for this comment.  In order to validate the consistency of the distribution from sub-domains characteristics with the Random Splitting module, we split a single source dataset into different sub-domains using the random sampling strategy. Then we use the pre-trained model to extract features from each sub-domain, and tSNE is applied to compare the features.
> > > > >
> > > > > The corresponding visualization results are shown in Fig. 9 in the Appendix of the revised version due to the page limitation.

---

### Decision · Program_Chairs · 2023-01-20

**Decision:**

Reject

**Justification For Why Not Higher Score:**

This paper receives 1x marginally below the acceptance threshold and 1x reject, not good enough.

**Justification For Why Not Lower Score:**

NA

**Metareview: Summary, Strengths And Weaknesses:**

This paper receives 1x marginally below the acceptance threshold, 1x accept, good paper, and 1x reject, not good enough. Overall, the ratings are leaning towards reject. The main criticisms are the presentation of the paper is not good. Many technical details are missing and notations are not used consistently. The experiments are lacking where evaluations are done only on two very old networks, PointNet and DGCNN. The proposed method does not outperform state-of-the-art (SLT, CVPR22). The performance is quite low compared with supervised learning approach, limiting the practicality of the proposed method.